# Do Quality Reviews Play a Role during the Implementation of Health Sciences Programmes? A Document Analysis Study

**DOI:** 10.3390/ijerph192113961

**Published:** 2022-10-27

**Authors:** Charity Ngoatle, Tebogo M. Mothiba, Modikana A. Ngoepe

**Affiliations:** 1Department of Nursing Sciences, University of Limpopo, Polokwane 0700, South Africa; 2Faculty of Health Sciences, University of Limpopo, Polokwane 0700, South Africa; 3Quality Assurance Unit, University of Limpopo, Polokwane 0700, South Africa

**Keywords:** quality reviews, implementation, teaching and learning, health sciences programmes, document analysis

## Abstract

Background: A programme review is a process that assesses the status, efficacy, and advancement of academic programmes and aids in determining their future needs, priorities, and direction. The purpose of the academic programme review is to demonstrate that the programmehas appropriate quality assurance processes and procedures in place in accordance with the applicable established criteria and to offer ongoing guidance for the development of academic programmes to ensure that they remain responsive and relevant. The study, therefore, sought to investigate the role of quality reviews during the implementation of health sciences programmes at a rural University in Limpopo Province, South Africa. Methods: Data were collected using the document analysis review technique to assess the Self-Evaluation Review reports for three programmes in the faculty of health sciences. The study’s descriptive qualitative data were analysed using a thematic analysis approach in six stages. All-inclusive purposive sampling was used to select the documents for review. Results: Three health sciences programmes were reviewed; two of the programmes met the minimum standards whereas one needed improvement. The review showed inadequate staffing, poor student support, and inadequate marketing of the programmes among others. Conclusions: The study has shown that conducting a review is crucial for maintaining and enhancing quality provisioning of programmes. The quality gaps identified by the panellists while reviewing the programmes can be used to improve and enhance quality of the programmes to a higher level if properly implemented. Thus, quality review does play a significant role during the implementation of health sciences programmes.

## 1. Introduction

Programme review is a process for assessing the status, efficacy, and advancement of academic programmes and assists in determining their future needs, priorities, and direction [1]. A Programme Review is carried out by both internal institution members and external peer reviewers. It consists of self-evaluation procedures that an institution undertakes to monitor and reflect on the outcomes and impact of its programmes. The primary purpose of the programme reviews is developmental in a sense that they afford an institution an opportunity to assess the quality of its programmes, facilitate due improvements and disseminate best practices where necessary to enhance the quality of the delivery of programmes [2].

The goals of the academic programme review are to provide ongoing improvement and enhancement of the quality of academic programmes [1], and to show that the programme has proper quality assurance systems, policies, and procedures in place [2]. The Council on Higher Education (CHE) [3] asserts that it remains the primary responsibility of Higher Education Institutions (HEIs) to ensure institutional and programme quality. Institutions should work to develop and maintain efficient systems and mechanisms that support the provision of high-quality programmes and produce accurate data for internal planning, self-evaluation, external evaluation, and public reporting [3].

Within a programme review, there is self-evaluation, which serves as the primary purpose of assessing the status of an academic programme and ultimately promoting the improvement of the quality of the programme. This is done by ensuring that the programme is meeting its objectives and has procedures in place to keep track of this, as well as that its quality assurance mechanisms, including itsstatements, are evidence-based [2].

In South Africa, the Council on Higher Education (CHE) is a quality assurance body for higher education programmes with its permanent quality assurance committee referred to as the Higher Education Quality Committee (HEQC). The HEQC’s duties include the national review of academic programmes [4]. The primary goals of a national review are to ensure that: minimum programme standards are met; students are protected from programmes that do not meet minimum quality standards; and that the public trust in higher education programmes is maintained [3].

In addition to the HEQC responsibility, higher education institutions are dutybound to manage the quality of their programmes internally. They do so through internal quality review processes. The most crucial document in the internal quality review process, as is the case with the national review process of programmes, is the Self-Evaluation Report (SER). It is the primary document developed by each participating academic division in an institution in accordance with the standards and minimum requirements premised on the Criteria for Academic Reviews and Manual for the Academic Review Process documents developed and approved by an institution [4].

The SER gives each department the opportunity to develop a critical evaluation of the programme premised on fair analysis and reflection with the goal of improving and enhancing the quality of the programme. Departments have the chance to use this SER document to pinpoint areas for best practices, those that need improvement, and other interventions to raise the programme’s overall quality [4].

The institution, through internal quality reviews as is the case with national reviews by the HEQC, also conducts site visits which is a crucial component of a programme’s internal review. The site visit for internal quality reviews is conducted by a peer review panel of internal and external professionals who assesses the SER and accompanying evidence. The panel corroborates evidence in the SER with the one gathered through interaction with programme stakeholders, interviews, and inspection of facilities during the site visit, and concludes whether the programme can be re-accredited. The degree to which the programme satisfies the predetermined criteria and minimum standards determines the programme’s status for re-accreditation [4].

The internal quality reviews criteria similar to the HEQC, and further pronounces that programmes must satisfy both the programme-level requirements and the national qualification standard to be re-accredited. Subsequently, prior to offering any programmes, HEIs is a requirement that must demonstrate their ability to do so while adhering to the standards and criteria intended to support such quality [3]. However, this study had the following question in mind: “*Do quality reviews play a role during the implementation of health sciences programmes*”? To answer the above question, the study, therefore, sought to investigate the role of quality reviews during the implementation of the health sciences programmes at the rural University in Limpopo Province, South Africa.

## 2. Methodology

### 2.1. Research Design

The study used a descriptive qualitative research design to describe the reports for the Faculty of Health Sciences; programmes reviewed against the University of Limpopo Criteria for Academic Reviews and Manual for the Review Process modelled on the Council on Higher Education’s (CHE) Criteria for Programme Accreditation [5].

### 2.2. Study Site

The study was conducted at the University of Limpopo, Health Sciences Faculty. The Faculty has two schools, namely, the School of Medicine and the School of Healthcare Sciences.

### 2.3. Population and Sampling

The study data sources included all the University of Limpopo, Faculty of Health Sciences programmes reviewed between 2019 and 2022 by the Council on Higher Education (CHE). The total population purposive sampling technique was used to include the three (3) SER documents in the Faculty of Health Sciences’ programmes that met the inclusion criteria [6].

#### Inclusion and Exclusion Criteria

The study included all the Self-Evaluation Reports (SER) prescribed by the Criteria for Academic Quality Reviews and Manual for the Academic Review in the University of Limpopo, for the internal quality review of the Faculty of Health Sciences’ programmes from 2019 to 2022. All documents designed for quality assurance by the external professional bodies for health sciences programmes were excluded.

### 2.4. Data Collection Procedures

Data were collected using the document analysis review technique [5].

### 2.5. Data Analysis

The study’s descriptive qualitative data were analysed using a deductive thematic analysis approach in six steps [5]. The steps include familiarising oneself with the data, generating initial codes, compiling codes with supporting data, grouping the codes into themes, reviewing and revising themes, and producing the report/writing the narrative [5].

### 2.6. Trustworthiness

Credibility was ensured through prolonged engagement with the data source, dependability was ensured through member checking, confirmability was ensured through peer review and availability of the data source, transferability was ensured through the dense description of the methodology, and authenticity was applied to ensure the rigor of the study [5].

## 3. Results

Three (3) health sciences programmes were reviewed between 2019 and 2022 based on the University of Limpopo Criteria for Academic Reviews and Manual for the Academic Review Process. The documents that were reviewed were for Master of Science in Medical Sciences, Master of Public Health (MPH), and Bachelor of Nursing Science (BNurs). The University of Limpopo Criteria for Academic Reviews, as modelled against and similar to the HEQC criteria for programme accreditation, consist of 16 criteria against which the programmes are measured. The panel consisted of eight (8) members as follows: two (2) external members from other institutions of higher learning, one (1) internal member from the institution Quality Assurance Unit, one (1) internal Quality Assurance Observer, and a report writer, three (3) internal members where one is the chairperson, and one (1) student representative from each programme. Each programme was reviewed against the criteria, and recommendations for each criterion were made.

**Exceeds Minimum Standard: EMS**, **Meets Minimum Standards: MMS**

**Needs Improvement: NI**, **Does not Comply: DNC**

### 3.1. Criterion 1: Programme Design

All three programmes were measured against this criterion (refer to Table 1) and recommendations were made as follows:

#### 3.1.1. Master of Science in Medical Sciences

The criterion was measured in conjunction with criterion 10: Programme review.

Recommendation—“*Implement mechanisms that would ensure that the student’s voice is heard in matters relating to feedback on the postgraduate student experience and decision-making in the programme*”.

#### 3.1.2. Master of Public Health

Recommendations—“*Refamiliarise staff members with SAQA programme level outcomes and UL approved programme design and module content and learning outcomes*”*. **Another recommendation** was*, *to* “*Benchmark with other institutions on how the programme is implemented to improve quality*”.

#### 3.1.3. Bachelor of Nursing Science (R425)

Recommendations were as follows: **firstly**, “*The philosophy of the curriculum must be clearly articulated and embraced by the Department*”. **Secondly**, “*Establish an advisory board for maintenance of the relationship between the university and the service platforms*”*. Thirdly*, “*Develop a phase-out plan for the current programme*”, **and lastly**, “*keep updated records of Evidence for South African Nursing Council (SANC) accreditation*”.

### 3.2. Criterion 2: Student Recruitment, Admission, and Selection

The recommendations for this criterion were recorded as follows:

#### 3.2.1. Master of Science in Medical Sciences

Three recommendations were: “*Find ways to streamline the student registration and re-registration process regarding the payment and reflection of registration fees*”. **Two**, “*Re-evaluate the marketing approach and recruitment drive to ensure programme marketing efficiency*”. **Number three was**, “*Identify broader ways of securing funding in a coordinated manner through research grant applications*”.

#### 3.2.2. Master of Public Health

Five recommendations emerged from the review as follows: **Firstly**, “*Present proposed admission criteria to the School and Faculty for discussion and approval*”. **Secondly**, “*Consult with Quality Assurance on admission criteria*”. **Thirdly**, “*Dispatch admission criteria through websites and marketing brochures*”. **The fourth one was**, “*Ensure that training opportunities in RPL are taken up by staff members*”, and **lastly**, “*Identify ways for improving recruitment strategy*”.

#### 3.2.3. Bachelor of Nursing Science (R425)

The programme reviews against this criterion for Bachelor of Nursing Science yielded eight recommendations from the panellists as follows: **number one is**, “*Efficiently market the programme to demonstrate to prospective students the opportunities resulting from completing the degree*”. **The second one is**, “*Explore the use of interviews*, *or other methods*, *as part of the selection to ensure that the correct student is admitted into the programme. In addition*, *the Department should be allowed to view and approved list of selected students prior to issuing acceptance letter*”. **Number three is**, “*Put mechanisms in place to address the attrition of the students who prefer to go to the college rather than the university due to financial support*”. **Number four is**, “*A tracer study should be done to determine the relationship between the admission score and the graduate throughput*”. **The fifth one is**, “*Benchmark with other Universities to ensure the AP score admission requirement provides for students to have an adequate chance for success in the programme*”. **Number six is**, “*Benchmark with other Universities on effective use of the online recruitment (admission and registration) system*, *to inform students timeously of admission*”. **Number seven being**, “*Establish systems and communication channels with the public to ensure that they are aware of the RPL access route*”. **Whereas**, **number eight is**, “*Consult with SANC on issues relating to the admission of disabled students*”.

### 3.3. Criterion 3: Staffing

The following recommendations were made per programme by the panellists for this criterion:

#### 3.3.1. Master of Science in Medical Sciences

This criterion was reviewed in conjunction with Criterion 13: Postgraduate Staffing. Two the recommendations were recorded as follows: **firstly**, “*the department must consolidate the staffing complement and plan for the renewal of temporary positions well ahead of time*”, **and secondly**, “*Meet with HR to put measures in place to ensure sufficiency of permanent staff in the existing staffing structure*”.

#### 3.3.2. Master of Public Health

Three recommendations emanated from the review: “*Recruit part-time & full-time staff and renew contracts before teaching activities to start*”. **Number two was recorded as**, “*motivates for new positions*”, **while number three was**, “*Explore ways to address supervision burden*”.

#### 3.3.3. Bachelor of Nursing Science (R425)

The review yielded six recommendations for the Bachelor of Nursing Science programme against this criterion as follows: **number one**, “*The University should investigate the ratio of permanent to part-time/Contract staff in the Department and ensure that this aligns to the minimum standards expected by the University and SANC*”. **Number two**, “*the department must introduce and manage the performance management system*”. **Number three is**, “*Workload models relevant for health science lecturers be implemented to provide objective evidence of workload and enable the HOD to do effective workload allocation*”. **Number four is**, “*When allocating more than one module per staff member*, *the Department should consider the implications of allocating modules at the same year level to one lecturer*”. **The fifth one is**, “*The University should ensure that a permanent post is created as part of the Departmental organogram for a clinical coordinator*”. **Number six is**, “*Consider using the clinical training grant funds to increase the number of administrators to at least two by appointing an administrator (fixed term to ensure consistency) that would address all clinical-related administrative matters*”.

### 3.4. Criterion 4: Programme Coordination

The recommendations for the three programmes against the programme coordination criterion were recorded as follows:

#### 3.4.1. Master of Science in Medical Sciences

This criterion was reviewed in conjunction with Criterion 12: Programme Coordination and Resources and Criterion 7: Resources, for this programme. Five recommendations were made as follows: **number one**, “*Leverage the realignment of human resources such as the linkage with the NHLS*”. **Number two**, “*Follow project appraisal processes by implementing feasibility checks and balances in terms of ensuring that financial and material capability is in place*”. **Number three**, “*Ensure the availability of monitoring procedures not only with Faculty*, *but also with the Procurement Division to process and follow-up with their suppliers and ensure that purchase requests relating to reagents and equipment for research are attended to as a matter of priority*”. **Number four**, “*Ensure that the ethical approval of projects take place in planned timeframes*”, **and number five as**, “*Clarify the respective roles of supervisor and co-supervisor during student induction and ensure the signing of a memorandum of understanding between student and supervisors*”.

#### 3.4.2. Master of Public Health

The following five recommendations were recorded for this programme: “Implement the programme as approved”. **Secondly**, “*Draw organogram incorporating all committees*”. **Thirdly**, “*Ensure adequate staffing*”. **Number four is**, “*Appoint HOD*”, **number five is**, “*Ensure research publication*”, **and lastly**, “*Contact with ASPHA/ PHASHA for affiliation*”.

#### 3.4.3. Bachelor of Nursing Science (R425)

The review yielded four recommendations for the Bachelor of Nursing Science programme against this criterion as follows: **number one**, “*Establish a mentoring and succession plan for various positions in the Department*, *especially that of the HoD position. Potential leaders should be identified and groomed to take leadership responsibility in the programme administration and coordination*”. **Number two is**, “*Establish a daily management committee to ensure the smooth running of the Department and programme*, *for example*, *checking all modules are on the timetable*, *and handling formative and summative assessment timetabling*”. **Number three being**, “*Where there is a turnover of administration staff*, *handover measures must be in place and implemented to ensure the smooth running of the Department*”, **whereas the fourth one was**, “*Ensure that agendas and minutes of meetings filed regularly*, *especially in the case of annual strategic plans and staff forum meetings*”.

### 3.5. Criterion 5: Teaching and Learning

The following recommendations were made in respect of the three programmes review:

#### 3.5.1. Master of Science in Medical Sciences

Criterion 5 was reviewed in conjunction with Criterion 11: Postgraduate Policies, Procedures, and Regulations, and two recommendations were made as follows: **number one**, “*Ensure that induction of students into postgraduate research is formally communicated and observed as per policy*”. **Number two**, “*Identify ways to acquaint supervisors with the benefits of the CPASA guide and attendant appendices and encourage supervisors to make use of the CPASA documents in the process of their supervision and monitoring of student progress*”.

#### 3.5.2. Master of Public Health

Only one recommendation was made for this programme as follows: “*Consider ways to clear the programme congestion*”.

#### 3.5.3. Bachelor of Nursing Science (R425)

Six recommendations were made as follows: **number one**, “*Reflect on*, *develop*, *and map student attributes related to the programme*, *and place into study guides and information books*”. **Number two**, “*Identify a unifying theoretical viewpoint that aligns with the programme purpose to act as a guiding principle in teaching the nursing modules*”. **Number three**, “*Develop a Teaching Philosophy for the programme and align the teaching model to the philosophy*”. **Number four**, “*Design templates for study guides module files to ensure uniformity*”. **Number five is**, “*Lecturer reviews should be done for theory lecturers in addition to the clinical lecturers. These must be done periodically*, *the data analysed*, *and feedback provided to the students*”. **Whereas number six is**, “*All lecturers need to be trained for the use*, *and the advanced use*, *of Blackboard in teaching and assessment*”.

### 3.6. Criterion 6: Student Assessment

The recommendations for criterion six were recorded per programme as follows:

#### 3.6.1. Master of Science in Medical sciences

The review for criterion 6 was done in conjunction with Criterion 15: Assessment and only one recommendation were recorded as follows: “*The school should ensure that an assessor database is in place to oversee assessor selection in harmony with the prescribed processes and of a transparent nature*”.

#### 3.6.2. Master of Public Health

Three recommendations emerged from the review against this criterion as follows: the **first one**, “*Staff should undertake refresher courses in assessment*”. **The second one is**, “*Provision of prompt feedback by staff*”, **and the last one is**, “*Moderate assessments*”.

#### 3.6.3. Bachelor of Nursing Science (R425)

The review against this criterion yielded three recommendations as follows: **number one is**, “*Open discussion of the memorandum and allow students access to the memorandum to check the marking*, *in addition to the current method of feedback being provided*”. **Number two is**, “*Develop a template that lecturers complete for every assessment where Bloom’s Taxonomy levels of the paper are indicated*”. **Number three is**, “*Student assessment results should be analysed statistically to show the mark distribution*”.

### 3.7. Criterion 7: Infrastructure and Resources

The recommendations for this criterion were recorded per programme as follows:

#### 3.7.1. Master of Science in Medical Sciences

The recommendations for the review against this criterion for the Master of Science in Medical Sciences programme were made in conjunction with criterion 4 and criterion 12. The recommendations were therefore outlined under criterion number 4 above.

#### 3.7.2. Master of Public Health

Four recommendations emerged for the review against this criterion as follows: **number one is**, “*Address the office space for staff members*, *and teaching and learning*”. **The second one is**, “*Address the OHS risks posed by the asbestos in the existing structures*”. **The third one is**, “*Encourage students to optimally utilize the library resources*”, **and number four is**, “*Utilise programme allocated library budget to the maximum*”.

#### 3.7.3. Bachelor of Nursing Science (R425)

Nine recommendations were made against this criterion when reviewing the BNurs programme as follows: **number one**, “*The University should consider planning for an academic building to be built that will satisfy the needs of the Nursing Department*, *and possibly consolidate the School of Health Care Sciences*”. **Number two**, “*The structural integrity of buildings should be maintained through regular maintenance*, *and occupational health and safety measures must be in place*, *such as access to emergency exits*”. **Number three**, “*More office space should be provisioned for staff in the nursing programme to ensure that all staff*, *inclusive of part-time staff have sufficient space to fulfil their functions*”. **Number four**, “*Part-time staff require access to the library and the Internet*”. **Number five**, “*Allocation of class venues for the programme should ensure sufficient space for all students.*
**Number six**, “*Optimal utilization of the allocated clinical training grant to purchase required consumables and appointment of staff for clinical teaching and learning*”. **Number seven**, “*Use of the skills laboratory to provide practical experience for isolated cases that requires specialized cases*, *which will relieve the pressure of sending students to training sites in Gauteng*”. **Number eight**, “*Consider investing in a larger skills laboratory for the shared use of all programme in the Health Sciences that require the use of a skills laboratory as part of their training*, *to allow for shared resources*, *and inter-professional teaching*”. **Number nine being**, “*Encourage the School of Medicine to urgently renovate their skills laboratory area to ease pressure on the nurses’ one in the interim*”.

### 3.8. Criterion 8: Co-Ordination of Experiential Learning

The recommendations for the reviews against this criterion were recorded per programme as follows:

#### 3.8.1. Master of Science in Medical Sciences

Not applicable.

#### 3.8.2. Master of Public Health

Not applicable.

#### 3.8.3. Bachelor of Nursing Science

Six recommendations emerged from the review against this criterion as follows: **number one**, “*A relevant clinical placement/WIL policy should be developed*, *approved*, *and implemented*”. **Number two is**, “*Ensure that students have indemnity cover*”. **Number three is**, “*Strengthen professional relationships with managers and staff in clinical practice through preceptor visibility*, *HOD engagement*, *and inclusion of clinical staff as advisory board members*”. **Number four is**, “*Develop clinical training model based on SANC guidelines and ensure active accompaniment of students*”. **Number five is**, “*More training sites in the regional and district hospitals must be accredited for decentralization of clinical training*”, **while number six is**, “*Develop a standard operating procedure for the management of health and safety incidences that may happen to students at clinical training sites*”.

### 3.9. Criterion 9: Student Retention, Student Throughput, and Programme Impact

The following recommendations were made against this criterion per programme:

#### 3.9.1. Master of Science in Medical Science

No recommendations were made.

#### 3.9.2. Master of Public Health

Two recommendations emerged from the review against this criterion for the Master of Public Health as follows: **number one**, “*Attend to all the causes that create obstacles relating to student retention and throughput rates*”. **Number two**, “*Conduct a study on the programme impact as part of its strategy to reposition the programme in SADC*”.

#### 3.9.3. Bachelor of Nursing Science (R425)

Two recommendations were also made for review of the Bachelor of Nursing Science against this criterion as follows: **number one is**, “*identify modules that students find difficult and provide additional support*”. **Number two**, “*Analyse students’ assessments to identify poor and good performing students*”.

### 3.10. Criterion 10: Programme Reviews

Recommendations were made for the review against criterion 10 per programme as follows:

#### 3.10.1. Master of Science in Medical Sciences

This criterion was discussed in conjunction with Criterion 1: Programme Design above.

#### 3.10.2. Master of Public Health

Three recommendations were made for the review against this criterion as follows: **number one**, “*Explore ways to use intermediary to conduct student surveys at the end of the module to allow students to freely express their comments*”. **Number two**, “*Discuss programme reviews in a formal meeting to improve the programme offering*”. **Number three is**, “*Implement students’ satisfactory surveys*”.

#### 3.10.3. Bachelor of Nursing Science (R425)

Three recommendations were made for the review against this criterion as follows: **first one**, “*Implement module reviews by students once per semester*”. **Secondly**, “*Provide students with career guidance to help them plan their career path*”. **Thirdly**, “*Review and reflect on the impact/effect of the programme for graduate employability regularly and make some proper alterations or adjustments*”.

### 3.11. Criterion 11: Postgraduate Policies, Procedures, and Regulations

Recommendations were made for the review against criterion 11 per programme as follows:

#### 3.11.1. Master of Science in Medical Sciences

This criterion was discussed in conjunction with Criterion 5: Teaching and Learning.

#### 3.11.2. Master of Public Health

One recommendation emerged from the review against this criterion as follows: “*Ensure availability of monitoring processes for effective implementation of postgraduate policies*, *procedures*, *and regulations*”.

#### 3.11.3. Bachelor of Nursing Science (R425)

Not applicable.

### 3.12. Criterion 12: Programme Coordination and Resources

Recommendations were made for the review against criterion 12 per programme as follows:

#### 3.12.1. Master of Science in Medical Sciences

This criterion was discussed in conjunction with Criterion: 4 Programme Coordination and Criterion: 7 Resources. It was outlined in criterion 4.

#### 3.12.2. Master of Public Health

The following recommendation was made for the review against this criterion: “*Utilise the available staff development resources and initiatives for professional development*”.

#### 3.12.3. Bachelor of Nursing Science (R425)

Not applicable.

### 3.13. Criterion 13: Postgraduate Staffing

Recommendations were made for the review against criterion 13 per programme as follows:

#### 3.13.1. Master of Science in Medical Sciences

This criterion was discussed in conjunction with Criterion: 3 Staffing above.

#### 3.13.2. Master of Public Health

One recommendation was made for the review against this criterion: “*Academic staff should obtain NRF rating as researchers*, *to improve understanding of research environment and how research impact on teaching and learning*, *whilst enhancing staff academic standing*”.

#### 3.13.3. Bachelor of Nursing Science (R425)

Not applicable.

### 3.14. Criterion 14: Student Development

The following recommendations were made for the review against criterion 14 per programme:

#### 3.14.1. Master of Science in Medical Sciences

Only one recommendation was made for the review against this criterion: “*Student supervisors must monitor and ensure the presence and development of graduate attributes of the students*, *especially those students engaged in retrospective research to ensure that they graduate with the required skills in place*”.

#### 3.14.2. Master of Public Health

Two recommendations were made for the review against this criterion: **number one**, “*Maximise the assistance offered by RDA for research training*”. **Number two**, “*Explore training opportunities offered by the library in research databases and referencing*”.

#### 3.14.3. Bachelor of Nursing Science (R425)

Not applicable.

### 3.15. Criterion 15: Assessment

The following recommendations were made for the review against criterion 15 per programme:

#### 3.15.1. Master of Science in Medical Sciences

This criterion was reviewed in conjunction with Criterion 6: Student Assessment above.

#### 3.15.2. Master of Public Health

Only one recommendation emerged from the review against this criterion as follows: “*Ensure availability of assessor database*”.

#### 3.15.3. Bachelor of Nursing Science (R425)

Not applicable.

### 3.16. Criterion 16: Research

The following recommendations were made for the review against criterion 16 per programme:

#### 3.16.1. Master of Science in Medical Sciences

Two recommendations were made for the review against this criterion as follows: **number one**, “*Consult with appropriate structures in the faculty and identify ways that will ensure postgraduate students would have a mechanism to table the problems they encounter during their research*”. **Number two**, “*Ensure that student perceptions on retrospective and prospective research design are well-informed perceptions as there is a tendency to regard one design to be superior to the other*”.

#### 3.16.2. Master of Public Health

One recommendation was made from the review against this criterion as follows: “*MPUC191 module should undergo review*, *to investigate the feasibility of including scientific writing*”.

#### 3.16.3. Bachelor of Nursing Science (R425)

Not applicable.

## 4. Discussion

### 4.1. The Document Review Has Revealed That the Designs for the Three Programmes Needed Some Improvement

For the Master of Science in Medical Sciences, the were no mechanisms in place to include students’ voices relating to students’ experience and decision making. A study conducted by Ngussa and Makewa [7] recommended that students should be allowed to participate in curriculum changes to empower and encourage them to take responsibility for matters that concern them. As a result, the inclusion of the students’ voices should be realized in all aspects of programme development [7].

For the Master of Public Health, the findings showed that the staff was unaware of the SAQA outcomes and the programme design that has been approved, and therefore were advised to benchmark with other institutions. According to Saldi [8] benchmarking will enable the staff to learn about other institutions’ best practices in governance, curriculum, continuous improvement, and resource management.

For the Bachelor of Nursing Science, the philosophy was not clearly embraced in the programme, there was no advisory board for maintenance of relationships with clinical institutions, no phase-out plan for the current programme (R425), and no updated record of SANC accreditation. Cheraghi, Yousefzadeh and Goodarzi [9] argue that philosophy establishes the foundation for decision making in all areas and determines the direction of activities. Similar findings were recorded where nurse educators were considered informed and helpful towards students learning; however, professional nurses felt that some educators did not communicate with the clinical staff sufficiently regarding student issues [10]. On the other hand, keeping a record of SANC accreditation is important because accreditation is a critical tool in making sure that programmes and degrees meet the highest educational standards [11].

### 4.2. The Findings Have Also Revealed That Student Recruitment, Admission, and Selection Need to Be Improved

There were poor marketing strategies for all three programmes. The students did not experience simplified payment and reflection of registration fees during registration processes, there was a lack of research grants and ineffective programme marketing strategies in the Master of Science in Medical Sciences programme. In the Master of Public Health programme, the results showed poor admission criteria and admission criteria communication, and poor RPL (recognition of prior learning) by staff members. Consequently, in the Bachelor of Nursing Science programme, the department was not involved in approving students before admission, there was no analysis of admission score versus throughput, the online recruitment system does not inform students of admission in a timely manner, and there were no SANC takes on the admission of disabled students into the programme. Similar findings have been documented by Tekieli Koay et al. [12] where analysis of individual academic programme websites was conducted to obtain pre-admission information and acceptance statistics from 260 graduate programmes in speech-language pathology accredited by the American Speech-Language-Hearing Association (ASHA). It was discovered that the information included on the individual websites and ASHA’s EdFind varied greatly between programmes, with some information being incomplete or not reported at all. This shows that online platforms should be used to effectively market the programmes by providing sufficient information pertaining to students’ recruitment, admission, and selection.

### 4.3. The Review Findings Showed That There Is Insufficient Staffing for the Programmes Which Resulted in an Increased Workload for the Available Staff

For the Master of Science in Medical Sciences and Master of Public Health programmes, the shortage of staff resulted in a supervision burden, while there was also a late renewal of temporary staff contracts. For the Bachelor of Nursing Science programme, the ratio of permanent to temporary staff needs to be investigated as there was allocation of more than one module at the same level to one lecturer, there was no performance management and development system (PMDS), and no clinical coordinator and clinical administrator to take care of clinical administrative issues. These findings are congruent to the study conducted by Engetou [13] indicating that staff shortages in an institution cause an increase in workload and less supervision. Many managers and supervisors believe that the level of an employee’s performance on the job is proportional to the position of the employee. However, Engetou [13] indicated that the workload is more about the work’s responsibilities, which is a common occurrence in many institutions worldwide. The workload in an institution normally occurs when employees undertake and carry out more tasks than is expected as per the workload model [13].

### 4.4. The Study Has Found That the Three Programmes’ Coordination Needs to Be Enhanced

In the Master of Science in Medical Sciences, there was no realignment of human resources, the project appraisal process was not followed and there was no monitoring procedures pertaining to procurement of resources, there was a prolonged ethical approval of projects, and the roles of supervisors and co-supervisors were not clarified with MOUs (memorandum of understanding).

For the Master of Public Health, the programme was not implemented as approved, the organogram did not incorporate all committees, there was inadequate staffing hampering with coordination of the programme, and the programme was not affiliated with ASOHA/ PHASHA. On the other hand, for the Bachelor of Nursing Science, there was no mentoring plan for staff, especially for the Head of Department (HoD) position, no daily management committee for the monitoring of the programme, and no handover measures in place for turnover of administration staff, and no filed agendas and minutes for meetings. Nabegu [14] indicates that academic institutions provide support services to guarantee the success of not only students but also the university’s specific mandate and the educational programmes’ goals. Support services commonly include, but are not restricted to, teaching information, available resources, level coordination, counselling, disability services, libraries, laboratories, and information technology.

### 4.5. The Results Show That the Teaching and Learning Functions of the Programmes Need Improvement

The Master of Science in Medical Science had poor communication of students’ induction into the postgraduate research as per the teaching and learning policy, and the supervisors were not accustomed to the CPASA guide. The Master of Public Health programme was congested. The Bachelor of Nursing Science’s student guides and information books did not have students’ attributions relating to the programme, there was no unified theoretical viewpoint that aligned with the programme’s purpose, no teaching philosophy for programme modules, no templates for unified study guides, no lecture reviews by students, and not all lecturers were trained to use Blackboard for teaching, learning, and assessment.

### 4.6. The Study Review Has Discovered That the Students’ Assessments in the Three Programmes Meet Minimum Standards

However, there was no assessor database to oversee assessor selection in the Master of Science in Medical Sciences, the staff needed a refresher course on assessment, there was no prompt feedback to students from staff, and assessments were not moderated. For the Bachelor of Nursing Sciences, the students did not have access to memoranda to check the marking, there was no specification table for assessments, and no statistical analysis of students’ results. Similar results were recorded in a study by Subheesh & Sethy [15] where the study emphasised the importance of assessment and feedback practices in achieving programme goals and objectives and ensuring programme quality. Assessment practices change over time in response to changes in programme course curriculum and instructional design in educational programmes. To deal with the ever-changing terrain of health sciences education, module course teachers must learn and implement effective assessment practices and types of feedback that motivate and inspire students to achieve modular course learning outcomes. However, most engineering faculty members around the world have little or no experience developing measurable course objectives, assessing student performance, and giving adequate and straightforward feedback to students [15]. As a result, Subheesh and Sethy [15] recommended that staff members seek help from educational experts to learn about efficient assessment and feedback strategies and encourage staff members from time to time to attend teacher-training programmes on instructional design, student assessment, and feedback practices.

### 4.7. The Results Have Revealed That There Is Inadequate Staff Office, Teaching, and Learning Space for the Master of Public Health and Bachelor of Nursing Science

The results have revealed that there were inadequate staff office, teaching, and learning space for the Master of Public Health and Bachelor of Nursing Science. There were occupational health risks and minimal utilization of the library budget allocated for the programme. The Bachelor of Nursing Sciences did not have a building that satisfied the needs of the programme, the current building lacked regular maintenance, part-time staff had no library access, the Clinical Training Grant from the Department of Higher Education and Training meant to supplement Council-controlled funding allocations due to the expensive training of students in health sciences programmes was not used efficiently and effectively, and the skills laboratory was not used for practical experiences requiring specialised cases which could save costs. There was no larger skills laboratory accommodating all health sciences programmes, and the skills laboratory for the MBChB programme was not renovated which created pressure on the Bachelor of Nursing Sciences skills laboratory. These findings are supported by Akomolafe and Adesua [16] where study findings revealed a significant relationship between the classroom environment and students’ academic performance. Therefore, institutions should, therefore, construct enough classrooms that are conducive to learning and renovate those that are in poor condition. Lecturers ought to create a pleasant and efficient learning environment in the classroom so that students can learn, perform better in class, and behave properly.

### 4.8. The Results Also Revealed That the Bachelor of Nursing Sciences’ Coordination of Experiential Learning Needs Improvement

The programme had no work-integrated learning (WIL)/clinical placement policy in place, students did not have indemnity cover, and there was a weak relationship between the programme and the clinical institutions. The clinical training model was not developed according to SANC guidelines, and there was no active student clinical accompaniment. Only training sites were accredited for clinical placement of students, and there were no standard operating procedures for managing health and safety incidences for students. The Ministry of Health in North Sydney [17] outlines the importance of a policy as laying out the steps that clinical institutions and Higher education institutions must take in order to facilitate clinical placements for students in health facilities and affiliated organizations. This includes creating a Student Placement Agreement, meeting specific compliance and verification requirements, and overseeing all clinical placements.

### 4.9. The Findings Have Revealed That There Are Hindrances Relating to Student Retention, Throughput, and the Programme Impact of the Master of Public Health and Bachelor of Nursing Sciences Programmes

The study on the programme impact was not conducted on the Master of Public Health. For the Bachelor of Nursing Sciences, students were not supported on the modules that they found difficult, and assessments were not analysed to identify poor and good-performing students. In a study, Passi, [18] argues that high levels of mentoring and student support should be maintained throughout the curriculum, by subject facilitators or supervisors. Consequently, the students must make the most of this assistance by using it to communicate any issues, ask questions, and seek professional guidance.

### 4.10. The Study Results Have Discovered That Programme Reviews on the Master of Public Health and Bachelor of Nursing Sciences Need Improvement

The study results revealed that the Criteria for Academic Reviews in relation to programme reviews on the Master of Public Health and Bachelor of Nursing Sciences needed review and improvement to accommodate other essential aspects of the two programmes not covered. It was discovered that student surveys were not done on both modules. The programme review was not addressed in formal meetings, which may distort programme offerings, and students’ satisfactory surveys were not implemented. For the Bachelor of Nursing Sciences, students were not provided with guidance to guide their career path, and regular review and reflection of the programme’s impact on the graduate’s employability was not done. Brereton, Schaefer, Bordilovskaya, and Reid [19] outlined the importance of using a survey for programme review and recommended that an online survey is conducted at the end of each semester to solicit student feedback on the course. All collected data are analysed by programme managers and used to inform the development of future programme iterations, such as changes to instructor training, assessment criteria, or course textbooks. Furthermore, lecturers receive all remarks written by their students, which can be used to inform future teaching practices.

### 4.11. The Results Showed That the Postgraduation Policies, Procedures, and Regulations Could Be Improved

For the Master of Public Health, there was no availability of monitoring processes for the effective implementation of postgraduation policies, procedures, and regulations. The University of Minnesota [20] highlights the importance of monitoring of policies, procedures, and regulations as crucial and states that monitoring entails ensuring that policies’ requirements are followed. The University of Minnesota further indicates that information obtained through monitoring gives responsible officers or their designees, frequently the policy owner, the chance to pinpoint issues with current policies’ compliance and address low compliance rates.

### 4.12. The Findings Also Revealed That the Programme Coordination and Recourses in Master of Public Health Programme Need to Be Improved as There Is Poor Staff Development

The findings also revealed that the programme coordination and recourses in Master of Public Health programme needed to be improved as there was insufficient staff development. The academic staff for the Master of Public Health was not NRF (National Research Foundation) rated.

### 4.13. The Findings Have Revealed That Postgraduate Students’ Development Could Be Improved

For Master of Science in Medical Sciences, supervisors did not monitor the presence and development of graduate attributes, particularly those students involved in retrospective research. In the Master of Public Health, the assistance offered by RDA was not maximally used for research activities, and library training opportunities for students were not explored on research databases and referencing. However, student support for developing the postgraduate candidate is crucial. A study conducted by Asamoah [21] showed that students benefited from an extensive institutional learner support system that included access to a learning management system, a computer laboratory, and software engineers who were occasionally available to assist with computer tool navigation. To assist them in using the manual and the online library resources, the students also had access to a qualified university librarian [21]. Furthermore, academic staff should play a key role in the implementation of graduate attributes in order to claim institutionally based attributes and apply them to fields of study [22].

### 4.14. The Findings on Study Results Showed That There Are a Few Aspects That Need Attention Relating to Research

There were no measures in place for students to report the challenges faced in their research, and students’ perceptions of retrospective and prospective research designs were not well-informed, as students tended to think that one was superior to the other in the Master of Science in Medical Sciences. For the Master of Public Health, the module MPUC191 needs feasibility investigation to include scientific writing. Qasem and Zayid [23] made several recommendations to help students overcome obstacles and succeed in the early stages of writing research proposals and projects. They [23] indicated that more emphasis should be placed on academic writing in English, with more activities, tasks, and training workshops to enhance research knowledge. Moreover, supervisors should provide sound advice to students.

## 5. Conclusions

The study examined the quality assurance review documents for the three Faculty of Health Sciences programmes for the 2019–2022 review period. According to the findings of the study, conducting a review is critical for developing, improving, and enhancing the quality of an academic programme. If properly implemented, the gaps identified by the panellists while reviewing the programmes can amplify the programmes to a higher level. Thus, quality review plays a significant role in the implementation health sciences programmes. The study findings are, however, limited to the implementation of health sciences programmes’ reviews reported in the context of the study, and therefore cannot be generalised to other programmes or settings. Consequently, for comparison purposes, the study recommends the conducting of similar studies in other settings or faculties. This could establish whether similar results would be yielded or not.

## Figures and Tables

**Table 1 ijerph-19-13961-t001:** Presents the results of the programme review per criterion.

CRITERIA	PANEL RATING PER PROGRAMME		
	MSc Med	MPH	BNURS
**Criterion 1: Programme Design**	-	MMS	MMS
**Criterion 2: Student Recruitment, Admission, and Selection**	MMS	NI	MMS
**Criterion 3: Staffing**	-	MMS	NI
**Criterion 4: Programme Coordination**	-	MMS	MMS
**Criterion 5: Teaching and Learning**	-	MMS	MMS
**Criterion 6: Student Assessment**	-	MMS	MMS
**Criterion 7: Infrastructure and Resources**	-	NI	NI
**Criterion 8: Co-ordination of experiential**	Do not apply	Do not apply	MMS
**Criterion 9: Student retention, student throughput, and programme impact**	-	NI	NI
**Criterion 10: Programme Reviews (in conjunction with** **Criterion 1)**	MMS	NI	NI
**Criterion 11: Postgraduate Policies, Procedures, and Regulations (in conjunction with Criterion 5 Teaching and Learning)**	MMS	MMS	Do not apply
**Criterion 12: Programme Coordination and Resources (in conjunction with Criterion 4 Programme Coordination and Criterion 7 Resources)**	NI	MMS	Do not apply
**Criterion 13: Postgraduate Staffing (in conjunction with** **Criterion 3 Staffing)**	NI	NI	Do not apply
**Criterion 14: Student Development**	MMS	NI	Do not apply
**Criterion 15: Assessment (in conjunction with Criterion 6** **Student Assessment)**	MMS	MMS	Do not apply
**Criterion 16: Research**	MMS	NI	Do not apply
**OVERALL RATING**	**MMS**	**NI**	MMS

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
