# Peer review of "Do Quality Reviews Play a Role during the Implementation of Health Sciences Programmes? A Document Analysis Study"

_ijerph, 2022, doi:10.3390/ijerph192113961_

Round 1

Reviewer 1 Report

Dear Authors,

First, I would like to congratulate you on the research presented.

After reading the article, I had some doubts, which I will now describe. 

In my opinion, it is not clear what the objectives or contributions of the work undertaken are. 

You could clarify the six steps mentioned in the methodology and link them to your study. 

The last section should include limitations and suggestions for future research. 

Another suggestion is about formatting. There are some errors in the use of abbreviations, for example, Higher Education Institutions (see lines 41, 70). Another example is Self-evaluation Review (see lines 76, 85, 88). Subheading 3.2.3 should be aligned with 3.2.2.

I wish you all the luck with your article.

Kind regards, 

Author Response

COMMENT: In my opinion, it is not clear what the objectives or contributions of the work undertaken are. RESPONSE: comment effected on line 20.

COMMENT: You could clarify the six steps mentioned in the methodology and link them to your study. RESPONSE: comment effected on lines 118 - 120.

COMMENT: The last section should include limitations and suggestions for future research. RESPONSE: comment effected on lines 677 - 680.

COMMENT: Another suggestion is about formatting. There are some errors in the use of abbreviations, for example, Higher Education Institutions (see lines 41, 70). RESPONSE: comment effected on line 47 and 89.

Another example is Self-evaluation Review (see lines 76, 85, 88). RESPONSE: comment effected on line 67. Subheading 3.2.3 should be aligned with 3.2.2. RESPONSE: comment effected on line 178.

Reviewer 2 Report

Dear authors,

1. My main concern is that the titel and -to some extent- the abstract do not correspond to the article, they are rather misleading. For example, the article does not answer the question in the title. I suggest a title like "The role of quality reviews in the implementation of health science programmes. A case study in South Africa". Please also rewrite the abstract and focus more on what you have actually done, not so much on the method / theoretical background.

2. Please add a paragraph to the introduction which describes the aim of the paper.

3. Please describe "CHE" at its first appearance in line 40 (rather tha in line 49).

4. What do you mean by "the size of the employee" (line 481)? It sounds like their hight, but this can't possibly be the case... Do you mean "size of the workforce"?

Author Response

COMMENT1. My main concern is that the titel and -to some extent- the abstract do not correspond to the article, they are rather misleading. For example, the article does not answer the question in the title. I suggest a title like "The role of quality reviews in the implementation of health science programmes. A case study in South Africa". Please also rewrite the abstract and focus more on what you have actually done, not so much on the method / theoretical background. RESPONSE: comment effected, check the abstract and line 682.

COMMENT: 2. Please add a paragraph to the introduction which describes the aim of the paper. RESPONSE: comment effected on lines 93-95.

COMMENT: 3. Please describe "CHE" at its first appearance in line 40 (rather tha in line 49). RESPONSE: comment effected on line 49.

COMMENT: 4. What do you mean by "the size of the employee" (line 481)? It sounds like their hight, but this can't possibly be the case... Do you mean "size of the workforce"? RESPPONSE: comment effected on lin 513.

Reviewer 3 Report

Dear Authors,

This is an interesting article that, I believe, contributes to the generation of academic knowledge.

I must congratulate the authors for the methodological clarity, from the type of study and methodology used, to the precise definition of the phases and measurement criteria used. 

This translates into the structure of the results, which are extremely detailed and well structured. Although, perhaps, in this section there is an abuse of short sentences that could perhaps be reordered or shown in a more graphic way indicator by indicator. The reader sometimes gets lost, and in others, the results are "short", or rather too briefly explained (For example: 3.1.2 What do these recommendations mean?).

However, the discussion conveniently reflects the analysis of the results, and is pertinent.

Perhaps the conclusion is short, and could be improved. It is very general, especially considering the quantity and quality of the results and their subsequent discussion. Can it be elucidated or further defined what the amplification of the programs to a higher level would mean? What does this mean for the authors?

Finally, the bibliography is more or less up to date, however, it could be somewhat more extensive.

Author Response

COMMENT: This translates into the structure of the results, which are extremely detailed and well structured. Although, perhaps, in this section there is an abuse of short sentences that could perhaps be reordered or shown in a more graphic way indicator by indicator. The reader sometimes gets lost, and in others, the results are "short", or rather too briefly explained (For example: 3.1.2 What do these recommendations mean?). RESPONSE: comment effected on line 162-163.

COMMENT: Perhaps the conclusion is short, and could be improved. It is very general, especially considering the quantity and quality of the results and their subsequent discussion. RESPONSE: comment effected under conclusion. Can it be elucidated or further defined what the amplification of the programs to a higher level would mean? What does this mean for the authors? RESPONSE: comment effected in line 681-683.

COMMENT: Finally, the bibliography is more or less up to date, however, it could be somewhat more extensive. RESPONSE: the comment is not clear.

Reviewer 4 Report

The paper deals with the review of three academic programs that are aimed at healthcare science. The execution of review is quite important from practical standpoint but it is questionable if this treatise represents meaningful scientific contribution.

 It is commonplace to identify research problem as well as the gap in the exploration of the problematics  in question. It was omitted.

As far as the goal is concerned I didn´t find any definition the goal of the paper . Setting the goal of the paper is quite importance for guiding both reviewers and readers through the paper. The problem might be superficial assessment of the programs that leads to suboptimum quality of its execution or something like this.

 I would appreciate e.g., the development of generic assessment methodology to be broadly applicable at universities or to identify barriers and obstacles to successful implementation of academic programmes.

 The methodology opts for qualitative approach based on the examination of written documents (thematic analysis as a subpart of content analysis). The quality of the program is then evaluated as per pre-defined qualitative criteria. In addition to missing goals (main and supporting goals) the paper is short of research questions (RQs) that draw attention to solution of specific scientific problems. It is not clear how the compliance with any individual criterion was measured. Who was in charge of the measurement (individual or the group of evaluators?). Where did the final judgement on meeting any criterion come from? What coding of the text did they use? Which themes did the authors generate out of codes?  Thermatic analysis can be either inductive or deductive. Which one did the authors use?

 Unfortunately, the paper suffers from many inconsistencies  both in structure and scientific content. The paper misses theoretical part that would address current level of knowledge in the assessment of academic programmes.

 The paper looks so simple and trivial and it is too far from having scientific value for  research and scientific community.

Author Response

 COMMENT: It is commonplace to identify research problem as well as the gap in the exploration of the problematics  in question. It was omitted. RESPONSE: comment effected in the previous reviewers comments.

COMMENT: As far as the goal is concerned I didn´t find any definition the goal of the paper . Setting the goal of the paper is quite importance for guiding both reviewers and readers through the paper. The problem might be superficial assessment of the programs that leads to suboptimum quality of its execution or something like this. RESPONSE: comment effected in the previous reviewers' comments.

 COMMENT: I would appreciate e.g., the development of generic assessment methodology to be broadly applicable at universities or to identify barriers and obstacles to successful implementation of academic programmes. RESPONSE: thanks for the suggestion, however, the paper has been restructured to suit the title.

 COMMENT: The methodology opts for qualitative approach based on the examination of written documents (thematic analysis as a subpart of content analysis). The quality of the program is then evaluated as per pre-defined qualitative criteria. In addition to missing goals (main and supporting goals) the paper is short of research questions (RQs) that draw attention to solution of specific scientific problems. It is not clear how the compliance with any individual criterion was measured. Who was in charge of the measurement (individual or the group of evaluators?). Where did the final judgement on meeting any criterion come from? What coding of the text did they use? Which themes did the authors generate out of codes? comment effected on lines 138-146. 

COMMENT: Thermatic analysis can be either inductive or deductive. Which one did the authors use? RESPONSE: comment effected on line 122.

 Unfortunately, the paper suffers from many inconsistencies  both in structure and scientific content. The paper misses theoretical part that would address current level of knowledge in the assessment of academic programmes.

Round 2

Reviewer 1 Report

Dear Authors,

Thank you very much for your answers and changes.

Best Regards,

Author Response

no comments were made.

Reviewer 4 Report

The authors responded to reservations and made some improvements. Now the paper slightly enhanced its quality and can provide reasonable information to academicians. That was one point that convinced me to support the paper for publication. I recomment adding a short outline of the further research in this field.

Author Response

COMMENT: I recomment adding a short outline of the further research in this field.

RESPONSE: The comment was addressed on page 17, line 688-690.